# Synthesis and Effect of Conformationally Locked Carbocyclic Guanine Nucleotides on Dynamin

**DOI:** 10.3390/biom12040584

**Published:** 2022-04-16

**Authors:** Kiran S. Toti, John R. Jimah, Veronica Salmaso, Jenny E. Hinshaw, Kenneth A. Jacobson

**Affiliations:** 1Laboratory of Bioorganic Chemistry, National Institute of Diabetes and Digestive and Kidney Diseases, NIH, Bethesda, MD 20892, USA; kiran.shambhu.toti@emory.edu (K.S.T.); veronica.salmaso@nih.gov (V.S.); 2Laboratory of Cell and Molecular Biology, National Institute of Diabetes and Digestive and Kidney Diseases, NIH, Bethesda, MD 20892, USA; john.jimah@nih.gov

**Keywords:** conformationally locked, methanocarba, guanine nucleotide, dynamin, GTPase, membrane fission

## Abstract

Guanine nucleotides can flip between a North and South conformation in the ribose moiety. To test the enzymatic activity of GTPases bound to nucleotides in the two conformations, we generated methanocarba guanine nucleotides in the North or South envelope conformations, i.e., (N)-GTP and (S)-GTP, respectively. With dynamin as a model system, we examined the effects of (N)-GTP and (S)-GTP on dynamin-mediated membrane constriction, an activity essential for endocytosis. Dynamin membrane constriction and fission activity are dependent on GTP binding and hydrolysis, but the effect of the conformational state of the GTP nucleotide on dynamin activity is not known. After reconstituting dynamin-mediated lipid tubulation and membrane constriction in vitro, we observed via cryo-electron microscopy (cryo-EM) that (N)-GTP, but not (S)-GTP, enables the constriction of dynamin-decorated lipid tubules. These findings suggest that the activity of dynamin is dependent on the conformational state of the GTP nucleotide. However, a survey of nucleotide ribose conformations associated with dynamin structures in nature shows almost exclusively the (S)-conformation. The explanation for this mismatch of (N) vs. (S) required for GTP analogues in a dynamin-mediated process will be addressed in future studies.

## 1. Introduction

One means of interrogating the mechanism of nucleotide-dependent processes is to use conformationally constrained bicyclic rings incorporated in the nucleotide analogues in place of ribose. Ribose may assume a range of ring twists, and if the protein-preferring conformation can be pre-established in the ribose-like ring, the potency and selectivity can be increased because of lowered energy barriers to binding. We have extensively explored the use of methanocarba (bicyclo[3.1.0]hexane) rings in this context [1,2,3,4]. By positioning the cyclopropyl ring that is fused to the cyclopentyl ring, either a North (N) or South (S) envelope conformation can be maintained. GTP-binding proteins and GTPases have widespread roles in receptor signaling and other biochemical processes in the cell. Although they are potentially useful probes of the biochemical role of guanine nucleotides, the (N) and (S) methanocarba analogues of GTP ((N)-GTP and (S)-GTP, respectively) have not previously been reported. We also introduce non-hydrolyzable modifications of the triphosphate moiety in the methanocarba-GTP series ((S)-GMPPCP), as was previously reported for a β,γ-methylene ATP analogue [4].

The effect of the novel methanocarba analogues of GTP ((N)-GTP and (S)-GTP) on GTPase activity was examined by using dynamin as a model system. Dynamin is a large GTPase that plays a role in numerous vesiculation events in the cell by wrapping around and pinching off budding vesicles. These membrane fission events include clathrin-mediated endocytosis, synaptic vesicle recycling, and intracellular vesical trafficking [5]. More recently, dynamin has been shown to bundle actin filaments prior to myoblast fusion, during muscle development [6]. The well-known membrane fission activity of dynamin is regulated by GTP binding and hydrolysis. Dynamin tubulates vesicles, constricts opposing lipid bilayers upon GTP binding (constricted state), and further constricts upon GTP hydrolysis (superconstricted state) that reaches the theoretical limit required for spontaneous membrane fission [7,8,9]. As the hydrolysis of GTP by dynamin is critical for its function, we examined how the conformational state of the GTP nucleotide affects dynamin membrane constriction and fission activity. The dynamin-bound GTP ribose ring appears to be in the (S) conformation according to a recent report [7].

In this study, we synthesized conformationally locked analogs: (N)-GTP, (S)-GTP, and (S)-GMPPCP. The effects of (N)-GTP and (S)-GTP on dynamin activity were probed by in vitro reconstitution of dynamin membrane constriction activity and imaging by cryo-electron microscopy (cryo-EM). Whereas dynamin tubulates liposomes and constricts the lipid tubules upon addition of GTP, only (N)-GTP, but not (S)-GTP, constricted lipid tubules. This suggests that (S)-GTP binding does not constrict lipid tubules or is markedly stalled in activity. We also examined dynamin structures in the protein databank and observed that they are predominantly in the (S)-conformation (stalled or stable state). This finding that (S)-conformation is stable (or stalled in activity) has implications for the structural biology of dynamin and other dynamin superfamily proteins to trap stable states of dynamin for X-ray crystallography and cryo-EM. The results also provide an avenue of investigation for future drug design and enzymology of GTPases.

## 2. Result and Discussion

### 2.1. Synthesis of Conformationally Locked Carbocyclic Guanine Nucleotides

We prepared three methanocarba-GTP analogues, i.e., **4a** (MRS4589, also referred to as (S)-GTP), **4b** (MRS4591, also referred to as (S)-GMPPCP), and **4d** (MRS4590, also referred to as (N)-GTP), for use in studying the conformational requirements of dynamin-GTPase (Figure 1), but a fourth planned analogue, **4c**, could not be synthesized. The ^1^H-NMR, ^31^P-NMR, and HRMS data are presented in the Appendix A. The key intermediate **2** was quickly accessed in moderate yields by reacting compound **1** [2,10] with 2-amino-4,6-dichloropyrimidin-*N*^5^-formamide under microwave conditions [11]. The treatment of this important intermediate **2** with aqueous Trifluoroacetic acid hydrolyzed the purine moiety and removed both protecting groups on the pseudo-ribose ring to give (S)-methanocarba-guanosine derivative **3** ((S)-mc-G). The conformationally locked guanosine was phosphorylated by using phosphorus oxychloride and a sequential addition of pyrophosphate gave (S)-mc-GTP **4a**. Unfortunately, the trimetaphosphate method, using salicylyl phosphorus (III) chloride, did not give **4a** [12]. Compound **4a** seemed to be unstable in neutral aqueous conditions or pure water/pure water, but it was stable in pH 7–8 triethylammonium acetate/ bicarbonate buffers. A similar approach to prepare **4b,c** did not materialize, and the reactions to synthesize (S)-mc-GMP [7] resulted in low yields. Hence, an alternate route was employed to access compound **7**. Sodium hydroxide–mediated hydrolysis gave a 2,3-isopropylidine protected guanosine analogue **5**, which was phosphorylated by using phosphoramidite chemistry to give **6**. A subsequent acidic hydrolysis using DOWEX-H^+^ resin gave the desired (S)-mc-GMP in excellent yields. Reacting the monophosphate **7** with tributylammonium salt of medronic acid in the presence of 1,2-dimethyl-*N*-tosylimidazolium triflate (TsDMIm-OTf) [13] gave β,γ-methylene-(S)-mc-GTP **4b**. It is perplexing that this method failed to give **4a,c** but was able to give (N)-mc-GTP **4d** [3]. Moreover, activation of the phosphate group of (S)-mc-GMP by using carbonyldiimidazole (CDI) and reacting with imidopyrophosphate did not give **4c**.

### 2.2. Effect of Conformationally Locked Carbocyclic Guanine Nucleotides on Dynamin Activity

We applied cryo-EM to investigate the effects of conformationally locked carbocyclic guanine nucleotides on dynamin activity. Consistent with previous reports, we observed via cryo-EM that dynamin alone tubulates liposomes and assembles as a helical polymer on lipid tubules with an outer diameter of ~50 nm (Figure 1A) [7,8,9,14,15]. Moreover, upon dynamin binding and hydrolysis of commercially available GTP, dynamin constricts the lipid tubule to an outer diameter of ~40 to 36 nm and results in fission (Figure 1B). Interestingly, we observed different effects of (N)-GTP and (S)-GTP on dynamin constriction and fission activity. Whereas the activity of dynamin upon binding to (N)-GTP was similar to the effect of GTP binding, binding to (S)-GTP was similar to the dynamin activity in the absence of nucleotide (Figure 1C,D).

In order to quantify the effects of conformationally locked carbocyclic guanine nucleotides on dynamin activity, we measured the diameter of dynamin-decorated lipid tubules in the presence or absence of various nucleotides. Briefly, we obtained cryo-EM micrographs of dynamin lipid tubules, similar to Figure 1, and boxed out segments of the tubes across many micrographs to obtain 5000 segments for each treatment: no nucleotide, GTP, (N)-GTP, and (S)-GTP. For each treatment, all the 5000 segments were overlaid in Fiji software (Figure 2A). Moreover, for each treatment, the 5000 segments were classified by 2D classification in RELION into 20 classes (Appendix A). Measurements of the outer diameters of the classes enabled us to compare the diameter of dynamin-decorated lipid tubules after various treatments (Figure 2B). We observed that treatments of GTP and (N)-GTP constricted dynamin tubes, while (S)-GTP failed to constrict, similar to the condition in the absence of GTP (Figure 2B).

We determined that (N)-GTP, but not (S)-GTP, enables dynamin-mediated membrane constriction. The North and South conformers of GTP ((N)-GTP and (S)-GTP) can be used with other GTPases in future studies, and it will be interesting to determine if this is a common phenomenon that (S)-GTP is not active. We hypothesize that (S)-GTP is inactive because the activity of GTPases requires transition through the North conformation.

Another explanation could be a defect of (S)-GTP in binding the GTPase, but this is likely not the case, as most structures of dynamin-related structures in the protein databank (Table 1) are bound to nucleotides in the South conformation (Figure 3). In fact, an ensemble of structures of the dynamin superfamily has been subjected to structural investigation to check the puckering propensity of nucleoside/nucleotide-like compounds bound to these targets. We computed the puckering parameters and the distribution of the pseudorotation phase angle (P), and of the degree of deformation from the plane (*ν*_max_) is reported in Figure 3 and Appendix A. Most of the nucleotide analogues fall in the South hemisphere of the polar plot, with the exception of GDP bound to the human OPA1 that assumes a N-E conformation, and of GDP and ADP bound to a bacterial dynamin-like protein (*Nostoc punctiforme*) and EH domain–containing protein 4 (*Mus musculus*), respectively, which are flat. Given the propensity of atomic structures of GTPases bind to GTP and its derivatives with a ribose (S)-conformation, we also hypothesize that (S)-GTP stabilizes the GTPases to enable their structural determination. Going forward, the new analogues will provide a powerful tool to explore the effects of methanocarba analogues (N)-GTP and (S)-GTP on a wide range of GTPases and to determine if the (S)-GTP conformation is predominantly inactive.

In conclusion, we synthesized the analogues of GTP containing a rigid ribose substitution, i.e., the methanocarba ring in both (N) and (S) fixed conformations. These correspond to two isomeric placements of a bicyclo[3.1.0]hexane ring system in place of the furanose ring. We used these analogues to interrogate the conformational requirements of a GTP-dependent dynamin-mediated process, i.e., membrane constriction and fission activity. Cryo-EM images of constricted dynamin-decorated lipid tubules show that (N)-GTP, but not (S)-GTP, enables this process. However, an automated survey of naturally occurring nucleotide ribose conformations in experimental structures of dynamin-related proteins shows almost exclusively the (S)-conformation. The explanation for this mismatch of conformations associated with dynamin-mediated processes, (N) in biochemical studies and (S) in experimental structures, will be addressed in future studies. We suggest that these new synthetic methanocarba–GTP analogues, including non-hydrolyzable modifications of the triphosphate moiety, here applied only to the dynamin system, will have broad applications as biochemical tools for studies of the role and conformation of guanine nucleotides in biochemistry [16].

## 3. Experimental

### 3.1. Chemical Synthesis

Compound **1** was synthesized according to the reported procedure [2]. *N*-(2-amino-4,6-dichloropyrimidin-5-yl)formamide was purchased from Combi-Blocks (San Diego, CA, USA). All other reagents and solvents were from Sigma-Aldrich (St. Louis, MO, USA). Anhydrous solvents were obtained directly from commercial sources. All reactions were carried out under nitrogen atmosphere, using anhydrous solvents. Room temperature or RT refers to 25 ± 2 °C. NMR spectra were recorded on a Bruker 400 MHz spectrometer. Chemical shifts are given in ppm (δ), calibrated to the residual solvent signals or TMS and frequency calibrated by solvent for ^31^P NMR (D_2_O δ = 4.79; *X*i = 40.480742, H_3_PO_4_ external; MestReNova 10.0.2). Exact mass measurements were performed on a mass spectrometer equipped with a standard electrospray ionization (ESI) and modular LockSpray TM interface. Ion-exchange chromatography (IE-LPLC) was performed by using a 10 × 100 mm Pharmacia column packed with GE Bioscience SOURCE 15Q resins, which was connected to Agilent 1100 HPLC system. The RP-HPLC was performed by using an (A) Agilent Eclipse XDB C18, 5 µm, 4.6 × 250 mm column or (B) Phenomenex Luna 5 µm C18(2)100A, AXIA, 21.2 × 250 mm column. Purity was determined by using either (A) or (C) Agilent ZORBAX SB-Aq, 5 µm, 4.6 × 150 mm column, at 1.0 mL/min linear gradient of acetonitrile/5 mM tetrabutylammonium dihydrogen phosphate (TBAP) as mobile phase. Purity of the final compounds as detected at 254 nm was >95%, unless specified otherwise. GTP was purchased from Jena Bioscience (Jena, Germany).

9-((3a*S*,3b*R*,4a*S*,5R,5a*R*)-5-(((tert-butyldiphenylsilyl)oxy)methyl)-2,2-dimethyltetrahydrocyclopropa[3,4]cyclopenta[1,2-d][1,3]dioxol-3b(3aH)-yl)-6-chloro-9H-purin-2-amine (**2**): A microwave reactor vial was charged with compound **1** (200 mg, 0.46 mmol), DIPEA (240 uL, 1.37 mmol), and *N*-(2-amino-4,6-dichloropyrimidin-5-yl)formamide (123 mg, 0.59 mmol) as a suspension in anhydrous n-butanol (4.0 mL). The sealed vial was irradiated to microwave for 3 h at 160 °C (Biotage, auto-power mode) [11]. The volatiles were rotary evaporated under vacuum, and the residue was purified by silica gel flash column chromatography to afford pure product **2** as a white solid (140 mg, 52%, *R*_f_ = 0.45; TLC eluent = 50% ethyl acetate in hexanes). ^1^H NMR (400 MHz, CDCl_3_) δ 7.76 (s, 1H, guanine 8-H), 7.74–7.66 (m, 4H, TBDPS Ph-H’s), 7.47–7.34 (m, 6H, TBDPS Ph-H’s), 4.94 (d, *J* = 7.0 Hz, 1H, 2′-H), 4.67–4.52 (m, 3H, 3′-H and guanine 2-NH_2_), 4.11–3.96 (m, 2H, 4′-C*H*_2_), 2.57 (t, *J* = 7.9 Hz, 1H, 4′-H), 1.94 (dd, *J* = 9.9, 5.5 Hz, 1H, 5′-H), 1.62 (t, *J* = 5.7 Hz, 1H, 5′,1′-c-Pr-H*_endo_*), 1.56 (s, 3H, 2′,3′-O-i-Pr), 1.41 (ddd, *J* = 10.0, 5.9, 1.6 Hz, 1H, 5′,1′-c-Pr-H*_exo_*), 1.22 (s, 3H, 2′,3′-O-i-Pr), 1.08 (s, 9H, TBDPS-*t*-Bu). ESI-HRMS [M + H]^+^ for C_31_H_36_N_5_O_3_SiCl calculated, 590.2354; found, 590.2353.

2-Amino-9-((1*R*,2*S*,3*R*,4*R*,5*S*)-2,3-dihydroxy-4-(hydroxymethyl)bicyclo[3.1.0]hexan-1-yl)-1,9-dihydro-6H-purin-6-one (**3**; (S)-mc-guanosine): A solution of compound **2** (15 mg, 0.025 mmol) in 70% TFA–water (2 mL) was stirred for 24 h. The solvents were rotary evaporated under high vacuum, and the residue was co-evaporated with ammonia in methanol. Purification by silica gel flash chromatography gave the title product **3** as a white solid (5.7 mg, 77%, *R*_f_ = 0.25; TLC eluent = 20% methanol in dichloromethane). ^1^H NMR (400 MHz, CD_3_OD) δ 7.72 (s, 1H, guanine 8-H), 4.57 (dd, *J* = 6.5, 1.8 Hz, 1H, 2-H), 4.04–3.92 (m, 2H, 3′-H, and 4′-CH_2_), 3.79 (dd, *J* = 11.4, 4.4 Hz, 1H, 4′-CH_2_), 2.25 (t, *J* = 3.9 Hz, 1H, 4′-H), 1.83 (t, *J* = 3.7 Hz, 2H, 5′-H, and 5′,1′-c-Pr-H*_endo_*), 1.19 (dt, *J* = 8.2, 6.2 Hz, 1H, 5′,1′-c-Pr-H*_exo_*). ESI-HRMS [M + H]^+^ for C_12_H_15_N_5_O_4_ calculated, 294.1202; found, 294.1201.

((1S,2R,3R,4S,5R)-5-(2-Amino-6-oxo-1,6-dihydro-9H-purin-9-yl)-3,4-dihydroxybicyclo[3.1.0]hexan-2-yl)methyl tetrahydrogen triphosphate (**4a**; (S)-mc-GTP) triethylamine salt: Lyophilized powder of (S)-mc-guanosine **3** (12 mg, 0.041 mmol) was added to a mixture of trimethyl phosphate (0.5 mL) and proton-sponge (26 mg, 0.123 mmol). The mixture was stirred at room temperature for 15 min and cooled to 0–5 °C, using an ice bath. Phosphorous oxychloride (11.3 µL, 0.123 mmol) was added, and stirring continued at 0–5 °C for 2–3 h. Then a mixture of tributylammonium pyrophosphate (225 mg, 0.41 mmol) and tributylamine (64 µL, 0.246 mmol) in anhydrous DMF (0.5 mL) was added at once, and the mixture was stirred at room temperature for 30 min. A cold 0.3 M triethylammonium bicarbonate solution (3 mL) was added, and stirring continued for 1 h. The reaction mixture was frozen and lyophilized. The residue was purified first by IE-LPLC (0 → 5 min, 100% H_2_O; 5 → 45 min, 100/0% to 0/100% H_2_O/1 M triethylammonium bicarbonate (TEAB) buffer, liner gradient at a flow rate of 2.0 mL/min, *R*t = 28.1 min), and then by C18-RPHPLC (column A, 10 mM triethylammonium acetate (TEAA) buffer/acetonitrile, 100/00 → 90/10 in 20 min, linear gradient at a flow rate of 1.0 mL/min, *R*t = 6.7 min) afforded the title compound **4a** as a white hygroscopic solid (1.35 mg, 4.5%). ^1^H NMR (400 MHz, D_2_O) δ 7.88 (s, 1H, 8-H), 4.31 (t, *J* = 6.3 Hz, 2H, 4′-CH_2_), 4.21 (d, *J* = 6.5 Hz, 1H, 3′-H), 2.46 (t, *J* = 6.5 Hz, 1H, 4′-H), 1.96 (d, *J* = 5.8 Hz, 1H, 5′-H), 1.76 (t, *J* = 5.3 Hz, 1H, 5′,1′-c-Pr-H*_endo_*). ^31^P NMR (162 MHz, D_2_O) δ −6.36, −10.86, −22.43. ESI-HRMS [M-H]^−^ for C_12_H_18_N_5_O_13_P_3_ calculated, 532.0036; found, 532.0040. Purity was 86.47%.

2-Amino-9-((3aS,3bR,4aS,5R,5aR)-5-(hydroxymethyl)-2,2-dimethyltetrahydrocyclopropa[3,4]cyclopenta[1,2-d][1,3]dioxol-3b(3aH)-yl)-1,9-dihydro-6H-purin-6-one (**5**): To a mixture of compound **2** (40 mg, 0.068 mmol) in dioxane (2 mL) was added 2 M NaOH (2 mL), and the mixture was heated to 100 °C for 3 h. The reaction mixture was cooled and neutralized with acetic acid. Volatile materials were evaporated under high vacuum, and purification of the residue by silica gel flash column chromatography gave the desired product **5** as a white solid (20 mg, 88%, R_f_ = 0.25; TLC eluent = 10% methanol in dichloromethane). ^1^H NMR (400 MHz, CD_3_OD) δ 7.73 (s, 1H, 8-H), 4.97 (dd, *J* = 6.9, 1.7 Hz, 1H, 2′-H), 4.74 (dd, *J* = 6.9, 1.4 Hz, 1H, 3′-H), 4.02 (dd, *J* = 11.3, 3.5 Hz, 1H, 4′-CH_2_), 3.85 (dd, *J* = 11.3, 4.4 Hz, 1H, 4′-CH_2_), 2.42 (t, *J* = 4.0 Hz, 1H, 4′-H), 2.02 (dd, *J* = 9.8, 5.4 Hz, 1H, 5′-H), 1.59 (t, *J* = 5.6 Hz, 1H, 5′,1′-c-Pr-H*_endo_*), 1.56 (s, 3H, 2′,3′-i-Pr-CH_3_), 1.45 (ddd, *J* = 9.8, 5.8, 1.8 Hz, 1H, 5′,1′-c-Pr-H*_exo_*), 1.29 (s, 3H, 2′,3′-O-i-Pr-CH_3_). ESI-HRMS [M + H]^+^ for C_15_H_19_N_5_O_4_ calculated, 334.1515; found, 334.1519.

((3aS,3bR,4aS,5R,5aR)-3b-(2-Amino-6-oxo-1,6-dihydro-9H-purin-9-yl)-2,2-dimethylhexahydrocyclopropa[3,4]cyclopenta[1,2-d][1,3]dioxol-5-yl)methyl di-tert-butyl phosphate (**6**): To a suspension of compound **5** (20 mg, 0.06 mmol) in THF (1 mL) was added tetrazole (40 mg, 0.54 mmol), followed by di-*tert*-butyl-*N*,*N*-diethylphosphoramidite (75 µL, 0.27 mmol), and the mixture was stirred at room temperature for 3 days. To this mixture at −78 °C was added a solution of mCPBA (~70%, 75 mg, 0.27 mmol) in dichloromethane (0.5 mL), followed by methanol (1 mL), and the mixture was stirred at room temperature of 30 min. Solvents were evaporated, and the residue was purified by silica gel flash chromatography to afford product **6** as a white solid (20 mg, 63%, *R*_f_ = 0.30; TLC eluent = 10% methanol in dichloromethane). ^1^H NMR (400 MHz, CDCl_3_) δ 9.09 (s, 1H, 1-NH), 7.76 (d, *J* = 5.0 Hz, 1H, 8-H), 6.83 (s, 2H, 2-NH_2_), 5.13 (d, *J* = 7.0 Hz, 1H, 2′-H), 4.65 (d, *J* = 7.1 Hz, 2H, 4′-CH_2_), 4.52 (s, 1H, 3′-H), 2.73 (t, *J* = 8.1 Hz, 1H, 4′-H), 2.21 (d, *J* = 13.4 Hz, 1H, 5′-H), 1.73 (t, *J* = 5.8 Hz, 1H, 5′,1′-c-Pr-H*_endo_*), 1.68 (s, 3H, 2′,3′-O-i-Pr-CH_3_), 1.62 (s, 9H, tBu), 1.58 (s, 9H, tBu), 1.36 (s, 3H, 2′,3′-O-i-Pr-CH_3_). ^31^P NMR (162 MHz, CDCl_3_) δ −9.64.

((1S,2R,3R,4S,5R)-5-(2-amino-6-oxo-1,6-dihydro-9H-purin-9-yl)-3,4-dihydroxybicyclo[3.1.0]hexan-2-yl)methyl dihydrogen phosphate (**7**; (S)-mc-guanosine) triethylamine salt: To a mixture of compound **6** (20 mg, 0.038 mmol) in 1:1 methanol-water (2 mL) was added DOWEX-H^+^ resin (150 mg), and the mixture was stirred at 60 °C for 4 h. the reaction mixture was cooled and neutralized with 0.3 M triethylammonium bicarbonate buffer (1 mL). The resin was filtered off, and the filtrate was lyophilized. The residue was dissolved in water and purified first by IE-LPLC (0 → 5 min, 100% H_2_O; 5 → 45 min, 100/0% to 0/100% H_2_O/1 M TEAB buffer, liner gradient at a flow rate of 2.0 mL/min, *R*t = 18.6 min) and then by C18-RPHPLC (column B, 10 mM triethylammonium acetate (TEAA) buffer/acetonitrile, 100/00 → 90/10 in 40 min, linear gradient at a flow rate of 5.0 mL/min, *R*t = 31.0 min) to afford pure product **7** as a white solid (17 mg, 94%). ^1^H NMR (400 MHz, D_2_O) δ 7.80 (s, 1H, 8-H), 4.60 (dd, *J* = 6.6, 1.9 Hz, 1H, 2′-H), 4.04 (t, *J* = 5.5 Hz, 2H, 4′-CH_2_), 3.99 (dt, *J* = 6.5, 1.2 Hz, 1H, 3′-H), 2.30 (t, *J* = 6.0 Hz, 1H, 4′-H), 1.83 (ddd, *J* = 9.5, 4.7, 1.4 Hz, 1H, 5′-H), 1.68–1.62 (m, 1H, 5′,1′-c-Pr-H*_endo_*), 1.27 (ddd, *J* = 9.4, 6.0, 2.0 Hz, 1H, 5′,1′-c-Pr-H*_exo_*). ^31^P NMR (162 MHz, D_2_O) δ 0.44. ESI-MS [M-H]^−^ for C_12_H_16_N_5_O_7_P calculated, 372.0709; found, 372.0716.

(((((((1S,2R,3R,4S,5R)-5-(2-Amino-6-oxo-1,6-dihydro-9H-purin-9-yl)-3,4-dihydroxybicyclo[3.1.0]hexan-2-yl)methoxy)(hydroxy)phosphoryl)oxy)(hydroxy)phosphoryl)methyl)phosphonic acid (**4b**, (S)-mc-β,γ-methyleneGTP) triethylamine salt:

Mixture A: To a solution of compound **7** (10 mg, 0.021 mmol) in anhydrous DMF (0.5 mL) was added 4Åå molecular sieves (25 mg) and the mixture stirred at room temperature for 1 h.

Mixture B: To a solution of bis[tetrabutylammonium] methylenediphosphonate (28 mg, 0.042 mmol) in anhydrous DMF (0.5 mL) was added 4Å molecular sieves (25 mg) and the mixture stirred at room temperature for 1 h.

To Mixture A at 0 °C was added TsDMImOTf [13] (12 mg, 0.0315 mmol). After stirring for 15 min, Mixture B was added, and then stirring continued for 1 h. the reaction was quenched by adding 0.3M TEAB buffer (1 mL), and the contents where lyophilized. The residue was purified first by IE-LPLC (0 → 5 min, 100% H_2_O; 5 → 45 min, 100/0% to 0/100% H_2_O/1 M TEAB buffer, liner gradient at a flow rate of 2.0 mL/min, *R*t = 27.4 min) and then by C18-RPHPLC (column B, 10 mM triethylammonium acetate (TEAA) buffer/acetonitrile, 100/00 → 90/15 in 40 min, linear gradient at a flow rate of 5.0 mL/min, *R*t = 26.7 min) to afford the desired product **4b** as a white hygroscopic solid (1.59 mg, 10.3%). ^1^H NMR (400 MHz, D_2_O) δ 7.79 (s, 1H, 8-H), 4.15 (t, *J* = 6.1 Hz, 2H, 4′-CH_2_), 4.06 (d, *J* = 6.6 Hz, 1H, 3′-H), 2.33 (t, *J* = 6.5 Hz, 1H, 4′-H), 2.22 (t, *J* = 20.4 Hz, 2H, β,γ-CH_2_), 1.88–1.75 (m, 1H, 5′-H), 1.65 (t, *J* = 5.5 Hz, 1H, 5′,1′-c-Pr-H*_endo_*), 1.27 (t, *J* = 7.9 Hz, 1H, 5′,1′-c-Pr-H*_exo_*). ^31^P NMR (162 MHz, D_2_O) δ 14.54 (d, *J* = 8.7 Hz), 8.23 (dd, *J* = 26.9, 9.0 Hz), −10.87 (d, *J* = 26.6 Hz). ESI-HRMS [M-H]^-^ for C_13_H_20_N_5_O_12_P_3_ calculated, 530.0243; found, 530.0250. Purity was 94.4%.

Preparation of bis[tetrabutylammonium]methylenediphosphonate: Medronic acid (1 eq) was dissolved in water and reacted with 1 M tetrabutylammonium hydroxide (2 eq). The mixture was frozen and lyophilized to obtain the compound as a white hygroscopic solid.

((1R,2R,3S,4R,5S)-4-(2-Amino-6-oxo-1,6-dihydro-9H-purin-9-yl)-2,3-dihydroxybicyclo[3.1.0]hexan-1-yl)methyl tetrahydrogen triphosphate triethylamine salt (**4d**, (N)-mc-GTP) [3]: Following the synthetic procedure described for **4b**, condensation of corresponding (N)-mc-GMP [3] (10 mg, 0.021 mmol) with tributylammonium pyrophosphate (23 mg, 0.042 mmol) and purification first by IE-LPLC (0 → 5 min, 100% H_2_O; 5 → 45 min, 100/0% to 0/100% H_2_O/1 M TEAB buffer, liner gradient at a flow rate of 2.0 mL/min, *Rt* = 35.2 min) and then by C18-RPHPLC (column B, 10 mM triethylammonium acetate (TEAA) buffer/acetonitrile, 100/00 → 90/10 in 40 min, linear gradient at a flow rate of 5.0 mL/min, *Rt* = 32.0 min) gave the title product **4d** as a white solid (1.47 mg, 9.5%). ^1^H NMR (400 MHz, D_2_O) δ 8.11 (s, 1H, 8-H), 4.89 (d, *J* = 6.8 Hz, 1H, 3′-H), 4.68 (s, 1H, 1′-H), 4.50 (dd, *J* = 11.2, 5.6 Hz, 1H, 4′-CH_2_), 4.07 (d, *J* = 6.8 Hz, 1H, 2′-H), 3.88 (dd, *J* = 11.2, 4.9 Hz, 1H, 4′-CH_2_), 1.81 (dd, *J* = 9.1, 4.0 Hz, 1H, 5′-H), 1.43 (t, *J* = 4.9 Hz, 1H, 4′,5′-c-Pr-H*_endo_*), 0.99 (d, *J* = 11.6 Hz, 1H, 4′,5′-c-Pr-H*_exo_*). ^31^P NMR (162 MHz, D_2_O) δ −6.39 (d, *J* = 21.2 Hz), −11.14 (d, *J* = 19.6 Hz), −22.55. ESI-HRMS [M-H]^−^ for C_12_H_18_N_5_O_13_P_3_ calculated, 532.0036; found, 532.0034. Purity was 94.8%.

### 3.2. Electron Microscopy of Dynamin-Mediated Constriction of Lipid Tubules

Full-length human dynamin 1 (dynamin) was expressed in HEK 293-F cells and purified by Ni-NTA affinity chromatography and gel filtration, as previously reported [6]. DOPS (1,2-dioleoyl-sn-glycero-3-phospho-L-serine) liposomes were produced by re-suspending dried DOPS lipid in HCB0 150 mM KCl buffer (20 mM HEPES pH 7.2, 1 mM MgCl_2_, 2 mM EGTA, 1 mM DTT, 150 mM KCl). The liposomes were extruded ~33 times through a 0.8 μm membrane, as previously described, to produce a 2 mg/mL DOPS liposome stock [7]. Dynamin-decorated lipid tubules were formed by incubating dynamin (6.5 μM final concentration) and DOPS liposomes (0.08 mg/mL final concentration) in HCB0 150 mM KCl buffer for 1 h. Then 4 μL of dynamin-decorated lipid tubules was applied to an electron microscopy grid (CF-1.2/1.3-4Au-50, Electron Microscopy Sciences, Hatfield, PA). After 30 s of incubation, 1 μL of HCB0 150 mM KCl buffer or 100 mM of nucleotide (GTP, N-GTP, and (S)-GTP) was applied to the grid and immediately plunge frozen in liquid ethane, using the EM grid plunger (Leica Microsystems, Morrisville, NC, USA). Electron micrographs of dynamin-decorated lipid tubules with or without nucleotide treatment were collected with a TF20 transmission electron microscope (FEI) at 200 kV and 29,000× magnification, using a K2 Summit camera (Gatan, Pleasanton, CA, USA). Cryo-EM image processing was performed with RELION 3.0 and Fiji.

### 3.3. Computation of Puckering Parameters

The computation of puckering parameters was performed by using an in-house script as a modified version of the reported script [17]. We used as input a list of UniProt [18] codes of protein of interests, as reported in the second column of Table 1. The script automatically retrieves the PDB and CIF files deposited in the Protein Data Bank [19] for each protein of interest. The SDF files of all the ligands in complex with the proteins (all chains) were downloaded, as well, and nucleoside/tide-like compounds were identified on the basis of SMARTS notation, as reported before [17]. The puckering parameters for these compounds were automatically computed and plotted on a polar plot reporting the pseudorotation phase angle (P) and the degree of deformation from the plane (*ν*_max_) on the angular and radial coordinates, respectively. The script was written in python3, using the modules *requests* and *urllib* for interaction with PDB and UniProt websites, *rdkit* [20] for SMARTS [21] substructure search and dihedral computation, numpy [22] and math for angles conversion and puckering computation, and matplotlib [23] for plotting purposes.

## Data Availability

The 2D classes of cryoEM micrographs of dynamin-decorated lipid tubules; representative GDP-bound dynamin (PDB ID: 2X2E) in South (S)-conformation with electron density shown around the ligand; ^1^H-NMR, ^31^P-NMR, HRMS, and HPLC data are presented in the Appendix A. Additional primary data are available upon reasonable email request.

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
