# Peer review of "Synthesis and Effect of Conformationally Locked Carbocyclic Guanine Nucleotides on Dynamin"

_biomolecules, 2022, doi:10.3390/biom12040584_

Round 1

Reviewer 1 Report

The chemical synthesis of nucleotide derivates is time demanding and challenging. Toti et al. describe in their manuscript the chemical synthesis of methanocarba analogues of GTP which allow for the systematic analysis of GTPases activity and their substrate binding specificity. To test the enzymatic activity of GTPases, they have chosen the model system dynamin.

Regarding the manuscript I have the following comments:

Strength of the manuscript include the synthesis of the methanocarba analogues of GTP. They also describe the challenges to synthesize non-hydrolysable derivates and apply CryoEM to analyze the effects of conformationally locked GTP on dynamin activity.

I really enjoyed the reading of the manuscript and have just minor comments:

Minor points:

  • Please give some information about the natural conformation of GTP in the introduction.
  • I do not understand the abbreviation Jena GTP? Please carify that “Jena” means.
  • MS spectra are completely missing. Please add them into the supporting information
  • Please add a reference to the second sentence of 2.2. “Consistent with previous reports…”.

Reviewer 2 Report

The manuscript written by Toti et al. combines the organic synthesis of two methanocarba guanine nucleotides (e.g., (N-) and (S)-GTP analogues) and their effect on dynamin activity. In this regard, the authors have studied the conformation effect of such GTP analogues on dynamin membrane constriction and fission activity using cryoEM images. The results presented are interesting, and the manuscript should be of interest to the readership of the Biomolecules. However, there is a couple of issues that prevent me from accepting this manuscript in the present form.

  • The authors should include in the supplementary part 1H-NMR, 13C-NMR spectra, and HR-MS of each compound that has been synthesized.
  • The chemical characterization is incomplete. The authors should add the 13C-NMR data of each compound in the experimental section.
  • The authors should assign as specifically as possible all of the carbon and proton environments in the spectra to the protons and carbons in the structure of all methanocarba guanine nucleotide compounds described in the experimental section.

Round 2

Reviewer 1 Report

The authors have improved the manuscript. However, the 1H, 31P, and HRMS data presentations need to be improved. The LC chromatograms are barely visible, and the peaks are not linked with the compounds. It would be great if the authors could provide the original electronic data and not only a scan of a printed paper. Moreover, I have not seen a link in the main text that guides the reader to the Supporting Information.
